# *Pitx*1 Enhancer Variants in Spined and Spine-Reduced Subarctic European Sticklebacks

**Dhurba Adhikari, Ida K. Hanssen, Steinar D. Johansen, Truls B. Moum and Jarle T. Nordeide ***

Genomic Division, Faculty of Biosciences and Aquaculture, Nord University, NO-8049 Bodø, Norway
* Correspondence: jarle.t.nordeide@nord.no; Tel.: +47-7551-7331

**Abstract:** Loss of body armour, sometimes including a reduction in or loss of pelvic spines, is an adaptation observed in many isolated freshwater populations. Pelvic reduction in sticklebacks has previously been associated with recurrent, but variant, deletions within pelvic enhancer regions *Pel*A and *Pel*B, which regulate expression of the homeodomain transcription factor gene *Pitx*1. We investigated variation in nucleotide sequences of pelvic enhancers in sticklebacks collected from two small freshwater lakes in the same watercourse and a nearby marine site in subarctic Norway. Spineless, as well as asymmetrically spined and completely spined sticklebacks are present in the upper lake, while only specimens with complete spines are found at the other lake and the marine site. Observed variation at *Pel*A between the three sites was mainly due to variable numbers of repeats at three fragile TG-repeat loci. The length of *Pel*A, mainly at one of the TG-repeat loci, was consistently shorter among individuals in the upper lake compared with specimens from the two other sites. However, no obvious association was revealed between enhancer variants and pelvic status. No polymorphism was found at *Pel*B. Thus, additional genetic factors and/or environmental cues need to be identified to fully explain the occurrence of pelvic reduction in sticklebacks in this lake.

**Keywords:** *Gasterosteus aculeatus*; stickleback; pelvic reduction; pelvic spines; TG-repeat; *Pitx*1; *Pel*A; *Pel*B; parallel evolution

**Key Contribution:** DNA sequence variation among sticklebacks suggests that additional genetic or environmental factors are involved in pelvic reduction than those shown by previous studies.





## 1. Introduction

Parallel phenotypic evolution has been defined as the independent evolution of the same trait in closely related lineages [1]. Parallel phenotypic evolution in organisms colonising new habitats may be due to either de novo mutations or standing genetic variation in the ancestral population (reviewed by [2]). Authors have advocated in favour of standing genetic variation as the most plausible mechanism due to its likely presence at higher frequencies, immediate availability in the new habitat, and because it has already been tested in similar environments [3–5]. Yet other studies support de novo mutations (reviewed by [6]).

Countless freshwater populations founded by marine ancestors after the last glacial period were trapped and isolated as land uplifted due to the deglaciation [7,8]. In addition, modern times human activity and perhaps birds might have transported species from saline to some freshwater habitats [8]. The threespine stickleback (*Gasterosteus aculeatus*) is one of the species with a marine origin that has colonized freshwater habitats, followed by physiological, behavioural, and morphological adaptations [9–17]. Marine threespine sticklebacks in general are protected against numerous piscivorous predators by strong external bony structures such as rows of lateral bony plates, pelvis structure (also termed pelvic girdle, which includes the pelvic spines), and dorsal spines [18,19]. A reduction in anti-predator armour, such as the lateral bony plates, may occur within a couple of decades of isolation in freshwater [20,21].

Dorsal and pelvic spines are assumed to give efficient protection against gape-limited predators such as fishes and birds, especially since the spines can be locked in the erect position [19]. Hence, the presence and length of the spines have been reported as positively associated with predation pressure from vertebrates [22–24]. However, in Cook Inlet, Alaska, freshwater populations of threespine sticklebacks with complete or partial loss of pelvic spines seem to be relatively abundant [12,18,25] (reviewed by [26]). A few pelvic reduced, freshwater populations have been reported elsewhere as well, e.g., from Western Canada [15,27–30], Iceland [28], Scotland [31], and Norway [16], reviewed by [26]. Such pelvic spine reduction may be selected for by invertebrate larvae, which are able to grab and hold on to the spines of juvenile spined sticklebacks [9] (but see [32,33]). Thus, low abundance of fish and bird predators and high abundance of insect predators could select for absence of spines or reduced spine length in sticklebacks, and vice versa. An alternative hypothesis to this "predation hypothesis" is the "calcium hypothesis", which advocates that low calcium ion concentration in freshwater could favour pelvic spine reduction [10]. Finally, the "predation-calcium hypothesis" argues that the combined effect of predators and low calcium ion concentration would be required to explain the evolution of pelvic reduction in sticklebacks [12].

A major determinant of pelvic development in threespine sticklebacks is the pituitary homeobox transcription factor gene *Pitx*1, located at chromosome 7 [15,28,31,34]. In addition, loci located at chromosome 2 [15], chromosome 4 [15,35], and chromosome 8 [29] have been suggested to play a role in fine-tuning of pelvic spine length. An enhancer element termed *Pel*A located upstream of *Pitx*1 (Figure 1) is reported as essential for the development of pelvic spines, and deletions at this locus have been shown to be associated with pelvic loss and reduction [28]. This *Pel*A pelvic limb enhancer is a cis-regulatory sequence, which contains multiple transcription factor binding sites, interacts with corresponding transcription factors, and enhances the transcription rate of *Pitx*1 [28,36]. Another enhancer element designated *Pel*B that maps downstream of *Pitx*1 (Figure 1) has been suggested to play a role in pelvic spine modification [37].

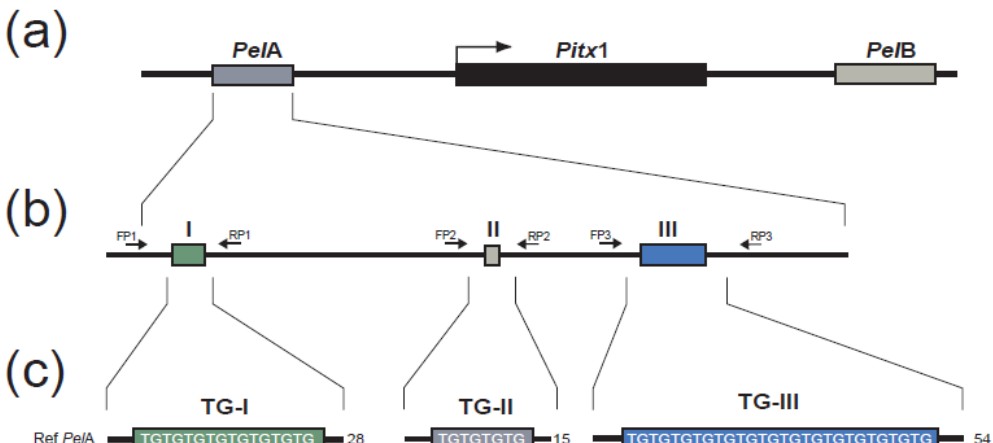

**Figure 1.** (**a**) *Pitx*1 with the upstream and downstream enhancers *Pel*A and *Pel*B, respectively. (**b**) *Pel*A with TG-repeats I, II, and III, and their relative location. FP1-3 and RP1-3 show the location of the forward and reverse primers used to sequence TG-repeat I, II, and III, respectively. (**c**) The reference sequence (SALR GU130435; 377,852 nt) is from a stickleback collected in Salmon River, British Columbia, and harbours 28, 15, and 54 TG-repeats at TG-repeat I, II, and III, respectively.

The *Pel*B enhancer was identified initially in mammals and is conserved between mice and fish including sticklebacks, in contrast to *Pel*A, which seems less conserved outside teleosts [37].

There are three TG-repeat arrays within the DNA-fragile region of *Pel*A, denoted TG-repeats I–III in the present paper (Figure 1), which likely contribute to deletion mutations

that are functionally related to pelvic reduction [28]. TG-repeats in the *Pel*A region may elicit a left-handed DNA helical structure, called Z-DNA [38–40]. This structure might affect the binding of transcription factors to the corresponding binding sites, causing an increase in the transcription rate [39]. Chan and colleagues [28] reported 9 different deletion patterns from 9 different spineless stickleback populations within the 2.5 kb *Pel*A region. These deletions are partially overlapping in a 488 bp region located at or near TG-repeats I–III [28]. The enhancer region's fragility and capability of forming a secondary DNA structure may explain the deletions of TG repeats I–III within the *Pel*A enhancer and the concomitant loss of pelvic spines in some threespine stickleback populations [39,40].

Studies in mammals have established the role of the *Pel*B enhancer as essential for pelvic hind limb development [37], but the corresponding biological role in sticklebacks is still a subject of interest and discussion. Spineless benthic sticklebacks from Paxton Lake in British Columbia have both a deletion of 125 bp and an insertion of 341 bp at *Pel*B, in addition to the mutations at *Pel*A (as discussed above) [37]. In addition, there might be other regulatory regions affecting pelvic development. For example, another transcriptional regulator, *Pitx*2, which is closely related to *Pitx*1, has been reported in vertebrates [41,42]. *Pitx*2 probably affects pelvis symmetry so that pelvic spines could be completely or partially lost at one side and less reduced at the other [41]. However, the role of the *Pitx*2 in pelvic spine reduction is not fully understood.

Pelvic reduction is reported from only 8 out of more than 200 Norwegian, mainly freshwater, populations examined [16,26]. In one of these lakes, Lake Storvatnet located in subarctic Northern Norway, 60% of the population lack one or both pelvic spines [43]. No pelvic reduction is observed in the downstream Lake Gjerdhaugvatnet in the same watercourse, or from a nearby marine site [43]. Specimens from both of the two freshwater populations have been categorized as "low plated" and marine specimens in this region have been categorized as "partially" and "completely" plated, based on the number of lateral bone plates [43]. Interestingly, Lake Storvatnet also contains an abundant population of brown trout (*Salmo trutta*) and identifiable stickleback parts were found in 19 per cent (N = 86) of the trout stomachs [43]. The abundance of insects, which may potentially prey on juvenile sticklebacks in Lake Storvatnet, was categorized as low [43]. A relatively large part of the population (≥30%) in Lake Storvatnet has grown 2 normal pelvic spines [43].

We studied the phenotypic variation of pelvic spines and the molecular variation at *Pel*A and *Pel*B in a comparison between (i) spined and spine-reduced sticklebacks within Lake Storvatnet, and (ii) Lake Storvatnet sticklebacks and (spined) conspecifics from the downstream Lake Gjerdhaugvatnet and marine specimens. Our hypothesis was that spineless sticklebacks from Lake Storvatnet have large parts of the *Pel*A enhancer deleted, similar to their North American spineless conspecifics [28]. We also hypothesised that more of *Pel*A was deleted in spineless compared with spined specimens in Lake Storvatnet and spined fish in the two nearby sites.

## 2. Materials and Methods

### 2.1. Sample Collection

A total of 427 sticklebacks were collected at three locations, 2 freshwater and 1 marine location, at Langøya island in Northern Norway in 2017, 2019, and 2020. The position (EU89 Lat/Lon), altitude, and size of the upper Lake Storvatnet are 68°46′49″ N, 15°9′36″ E, 80 m, and 0.2 km$^2$, respectively (Figure 2). Lake Gjerdhaugvatnet is a small lake (0.01 km$^2$) located downstream in the same watercourse at 20 m altitude. The two lakes are connected by an approximately 500 m brook with several waterfalls, which most likely prevent any gene flow between stickleback populations inhabiting the two lakes. Sticklebacks were sampled from marine or brackish water in the tidal mouth of a small river at Sandstrand (68°44′45″ N, 15°20′42″ E), here referred to as the marine site. The marine site is located about 8 km (direct distance) from the two other sampling sites (Figure 2). Sampling was carried out in June 2017, 2019, and 2020 in Lake Storvatnet, in June 2019 and 2020 in Lake Gjerdhaugvatnet, and in June 2020 at the marine site. Traps were deployed at 0.3–1.0 m

depth along the shore and retrieved about 24 h later. The sticklebacks were euthanised and sacrificed by an overdose of tricaine methanesulfonate (MS222) and then rinsed with water. The total length (from head to the posterior part of the caudal fin) of the body was measured by a ruler to the nearest mm. The caudal fin was cut off and discarded. Samples of the posterior fin muscle (about 5 mm in size) were homogenized immediately by bead beating using a Dremel 8220 rotary tool (MP Biomedicals, Solon, OH, USA) and 0.5 mL DNA/RNA Shield solution (Zymo Research, Irvine, CA, USA) and kept at low temperatures before further analyses at the laboratory. Specimens with body size less than 30 mm were discarded.

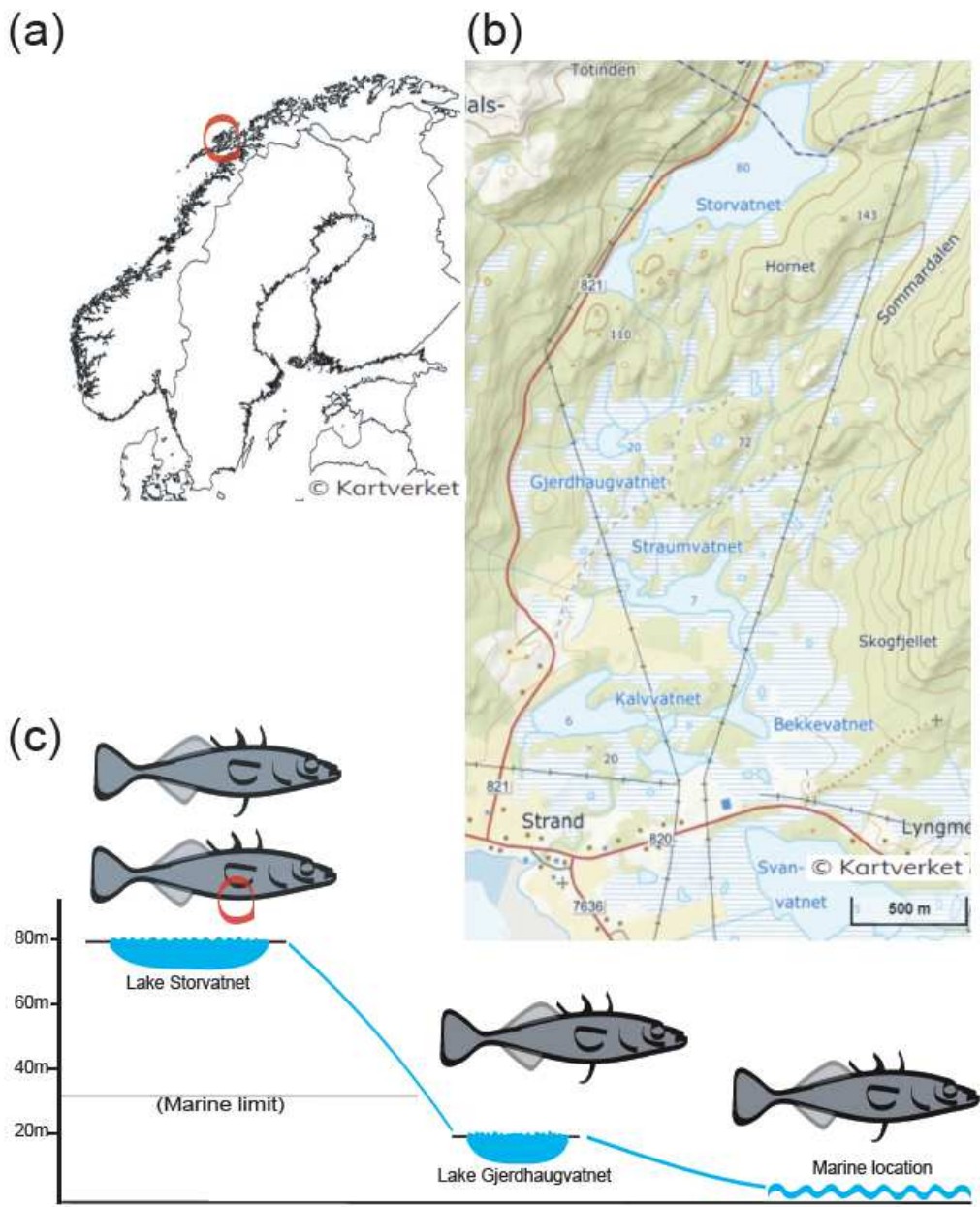

**Figure 2.** (**a**) Scandinavia with the study site encircled in red. (**b**) Map ([www.norgeskart.no](www.norgeskart.no) accessed on 28 March 2023) of the upper Lake Storvatnet (68°46′49′′ N, 15°9′36′′ E) and the lower Lake Gjerdhaugvatnet in the watercourse, whereas the marine sampling site (not shown) is located about 8 km from the two lakes. (**c**) Schematic drawing (out of scale) of the three sites showing the symmetric-spined, asymmetric-spined, and the spineless (encircled in red) sticklebacks in Lake Storvatnet. All specimens are symmetric spined at the two other sites.

To measure the $Ca^{2+}$ concentration in Lake Storvatnet and Lake Gjerdhaugvatnet, water samples were collected on 26 June 2021 from about 10 cm depth and about 1 m from land. The water samples were analysed by Labora AS (Bodø, Norway).

### 2.2. Morphology and Computation of Pelvic Scores

The specimens and their spines, and the tube where each specimen was stored individually, were examined for potentially broken spines. No broken spines were detected. The lengths of the right and left pelvic spines were measured by a digital calliper and a pelvic score (PS) of 0–4 was given to each side of the pelvis. Each side of a complete pelvis consists of an anterior process (ap), an ascending branch (ab), a posterior process (pp), and a pelvic spine (ps). PS 0 is for fish with no pelvic structure at all, PS 1 is for fish with ap only, PS 2 is for fish with ap + ab, PS 3 is for fish with ap + ab + pp, and PS 4 is for fish with a complete pelvis structure (ap + ab + pp + ps). A combined pelvic score (CPS) of 0–8 was assigned to each specimen by adding up the PS of both sides of the pelvis [12] (Supplementary Figure S1). CPS 0 is for fish with no pelvic structure at all, and CPS 8 is for fish with complete pelvic structure including pelvic spines [12]. The individuals were observed under a stereomicroscope ($10\times$ or $20\times$ magnifying lens) with gentle pressure on the pelvis by forceps to categorize PS. Samples were divided into three classes: (a) spineless, (b) symmetric spined, and (c) asymmetric-spined specimens (see details in the Supplementary Figure S1). The pelvis was defined as asymmetric if the difference between the length of the 2 pelvic spines was > 0.2 mm.

### 2.3. DNA Sequencing and Fragment Analysis

Muscle tissue for DNA analyses was taken from a total of 19 specimens from Lake Storvatnet. Seven symmetric-spined, six asymmetric-spined, and six spineless specimens were picked at random after categorising the sticklebacks into the three groups. Moreover, muscle tissue was sampled from 12 random specimens from Lake Gjerdhaugvatnet and 7 from the marine site. Genomic DNA was extracted from these 38 muscle tissue samples using the Monarch genomic DNA purification kit (New England Biolabs). The quality and concentration of DNA were checked with NanoDrop (Thermo Fisher Scientific, Waltham, MA, USA) spectrophotometry. DNA samples included in the study had a concentration of $\geq$ 20 ng/μL and absorbance ratios A260/A280 = 1.80–1.90 and A260/A230 = 1.80–2.50. All amplicons used for DNA sequencing and fragment analyses were produced with LongAmp Taq polymerase (Thermo Fisher Scientific, Waltham, MA, USA).

#### 2.3.1. *Pel*A Sequence Analyses

The genomic sequence of a *Pitx*1 allele from a marine pelvic-complete stickleback from Salmon River, British Columbia, was adopted as the reference sequence for the present study (Genbank accession no. GU130435 (377,852 nt)). Primers were designed based on the reference sequence using the "primer design tools" and "oligo analysis tools" of Eurofins Genomics (https://eurofinsgenomics.eu/) and named according to the position of their 3′ nucleotide in the reference sequence. DNA samples were amplified with forward and reverse primers: 5′-GCC CAA AAC TGA CAA AGC A-3′ (F128812) and 5′-AGC AGC AAA AGC AAA ATG AGA-3′ (R131624) targeting a 2813 bp region containing *Pel*A (*Pel*A amplicon) according to the reference. PCR was conducted with an initial denaturation at 94 °C for 90 s; 30 cycles of denaturation at 94 °C for 30 s; annealing at 59 °C for 20 s; extension at 65 °C for 150 s; and final extension at 65 °C for 10 min. PCR products were inspected by agarose gel electrophoresis, cleaned by ExoSap IT (Thermo Fisher Scientific, Waltham, MA, USA) according to the manufacturer's protocol, and subjected to direct sequencing with BigDye v3.1 (Applied Biosystems, Thermo Fisher Scientific, Waltham, MA, USA) following the manufacturer's instructions. Each of the segments containing TG-repeat I, II, and III, were sequenced in the forward and reverse directions using the following sequencing primers: 5′-AGG TCC ACA GTA CAG TGC AG-3′ (F128968) (FP1 in Figure 1) and 5′-TGG GAC GAG AAG ATG CCT TCA G-3′ (R129360) (RP1 in Figure 1),

5′-GTC GAA GCA AAG AGG CGA GAC ATC-3′ (F129687) (FP2 in Figure 1) and 5′-TTC TAA AGT GGT CGC TCG GC-3′ (R129962) (RP2 in Figure 1), and 5′-GTT ATG AAG GGC CGA GCG AC-3′ (F129933) (FP3 in Figure 1) and 5′-GCG TGA CCA CAA CAA TCC G-3′ (R130252) (RP3 in Figure 1) (Supplementary Figure S2). Sequencing reactions were treated with magnetic bind and ethyl alcohol, eluted with elution buffer, and run on a 3500xL Genetic Analyzer (Applied Biosystems) following the manufacturer's instructions. Sequencing results were analysed with the help of Finch TV version 1.4.0.

### 2.3.2. *Pel*A Fragment Analyses

The allelic length variation of the TG-repeats in the *Pel*A region was determined using fragment analyses. All three TG-repeats were amplified from DNA samples with primers corresponding to those applied in sequencing, and the forward primers to identify TG-repeat I, TG-repeat II, and TG-repeat III, were fluorescently labelled with FAM, FAM, and ATT056, respectively. Then, amplicons were diluted to 1:200 and treated with HiDi formamide (Thermo Fisher Scientific, Waltham, MA, USA) and run with standard ladder GeneScan 500 LIZ (Thermo Fisher Scientific, Waltham, MA, USA) on the 3500xL Genetic Analyzer. Fragment data were analysed with the software GeneMarker version 2.6.3. The exact number of TG-repeats for each allele was inferred, based on DNA sequencing and fragment analysis in combination. No conflict was revealed between the Sanger sequencing and the fragment analysis data.

### 2.3.3. *Pel*B Sequence Analyses

The *Pel*B region was studied for 10, 6, and 5 specimens from Lake Storvatnet, Lake Gjerdhaugvatnet, and from the marine site, respectively. DNA samples were amplified with a forward and a reverse primer: 5′-CAC GGA TTA CTG AGC AGC AA-3′ (F176680) and 5′-AGC TCA AGA CCT CTG GAT GG-3′ (R177688), targeting a 1009 bp region that harbours *Pel*B. PCR conditions consisted of an initial denaturation at 94 °C for 90 s; 25 cycles of denaturation at 94 °C for 30 s; annealing at 59 °C for 20 s; extension at 65 °C for 90 s; and final extension at 65 °C for 10 min. A 671 bp segment of the amplicon where length polymorphism was previously reported by [37] was sequenced in both directions as detailed above, using primers 5′-ACA GAC AGA CAG ACA GAC AG-3′ (F176836) and 5′-TAT ATC AAT CGA GAG AGG AAG AGG-3′ (R177550).

### 2.3.4. Identification of Single Nucleotide Polymorphisms (SNPs)

Successfully retrieved sequences from all specimens, including the upstream and downstream flanking sequences of TG-repeats I–III of the *Pel*A region, as well as *Pel*B sequences, were aligned to the reference sequence (GU130435) and their SNPs identified. Alignment was conducted with the help of Clustal Omega tools (https://www.ebi.ac.uk/Tools/msa/clustalo/).

The study was carried out according to ethical guidelines stated by the Norwegian Ministry of Agriculture and Food through the Animal Welfare Act. According to these guidelines, we were not required to—and therefore do not—have a specific approval or approval number.

## 3. Results

### 3.1. Morphology, Pelvic Scores, and Ca²⁺ Concentration

Descriptive statistics of body length and length of the pelvic spines of sticklebacks from Lake Storvatnet, Lake Gjerdhaugvatnet, and the marine site, are presented in Table 1. Of the 304 specimens from Lake Storvatnet, 113 (37%) were symmetric spined, 99 (33%) were asymmetric spined, and 92 (30%) were spineless (Table 1). The polymorphic stick-lebacks in Lake Storvatnet were classified into eleven groups based on their PS and CPS scores (Table 2). Note that none of these specimens had a CPS of 0 which means that none lacked the entire pelvic girdle (Table 2, Supplementary Figure S1). Among the asymmetric-spined sticklebacks from Lake Storvatnet, 29 had right-biased asymmetric

pelvic spines, and 70 had left-biased asymmetric pelvic spines (Table 1). All specimens collected from Lake Gjerdhaugvatnet (N = 73) and from the marine site (N = 50) were fully spined (CPS = 8) and symmetric (Table 1). Of the asymmetric-spined fish, 29 and 70 were right- and left-biased, respectively (Table 1), which is significantly different from unity ($\chi^2$ = 16.9, $p < 0.001$, d.f. = 1, chi-square test). Moreover, after including the one right-biased asymmetric spineless individual (Table 2) "P.v. j", the difference is significant ($\chi^2$ = 16.0, $p < 0.001$, d.f. = 1).

**Table 1.** Morphological measurements of threespine sticklebacks from the two freshwater lakes, Lake Storvatnet and Lake Gjerdhaugvatnet, and a marine site. The mean of pelvic spine lengths from Lake Storvatnet was calculated based on specimens with spines and asymmetric-spined specimens only. Pelvic scores (PS) were calculated for both the left and right side of the specimens and vary from 0–4. The combined pelvic score (CPS) is the sum of the PS from both sides and varies from 0–8. "N" is the total number of specimens from each location.

| Site | N | Spineless | Symmetric Spined | Asymmetric | | Spine Length (cm) (Mean ± Sd) | Body Length (cm) (Mean ± Sd) |
|---|---|---|---|---|---|---|---|
| | | | | Right-Biased | Left-Biased | | |
| Storvatnet | 304 | 92 (30%) | 113 (37%) | 29 (10%) | 70 (23%) | 0.26 ± 0.100 | 4.7 ± 0.60 |
| Gjerdhaugvatnet | 73 | 0 | 73 | 0 | 0 | 0.37 ± 0.070 | 4.1 ± 0.60 |
| Marine | 50 | 0 | 50 | 0 | 0 | 0.55 ± 0.100 | 4.8 ± 0.70 |

**Table 2.** Number of morphological variants from the three examined populations, Lake Storvatnet, Lake Gjerdhaugvatnet and the marine site. See also Supplementary Figure S1. "n" is the number of each morphological variant from each location. [1] The left column ("P.v.") refers to the pelvic spine morphs as shown in the Supplementary Figure S1.

| P.v. [1] | Pelvic Scores | | Combined Pelvic Scores (CPS) | Locations (n) | | | Remarks |
|---|---|---|---|---|---|---|---|
| | Left PS | Right PS | | Storvatn | Gjerdhaugvatn | Marine | |
| b | 4 | 4 | 8 | 113 | 73 | 50 | Symmetric spined |
| c | 4 | 4 short | 8 | 35 | 0 | 0 | Left-biased asymmetry |
| d | 4 short | 4 | 8 | 22 | 0 | 0 | Right-biased asymmetry |
| f | 3 | 4 | 7 | 7 | 0 | 0 | Right-biased asymmetry |
| e | 4 | 3 | 7 | 29 | 0 | 0 | Left-biased asymmetry |
| h | 4 | 1 | 5 | 5 | 0 | 0 | Left-biased asymmetry |
| g | 4 | 2 | 6 | 1 | 0 | 0 | Left-biased asymmetry |
| i | 3 | 3 | 6 | 32 | 0 | 0 | Spineless |
| j | 1 | 3 | 4 | 1 | 0 | 0 | Spineless |
| l | 1 | 1 | 2 | 51 | 0 | 0 | Spineless |
| k | 2 | 2 | 4 | 8 | 0 | 0 | Spineless |

The measured $Ca^{2+}$ concentration in Lake Storvatnet and Lake Gjerdhaugvatnet was 0.90 mg/L and 0.84 mg/L, respectively.

### 3.2. Allelic Variation of PelA

Allelic variation of *Pel*A was primarily caused by variable numbers of TG dinucleotides at three TG-repeat arrays. The allelic variation found in Lake Storvatnet differed strikingly from that of the two other sampling sites. TG-repeats I and III in particular, showed a wide range of length variants, with generally shorter arrays found in individuals from Lake Storvatnet. At TG-repeat III, the predominant allele in Lake Storvatnet was $(TG)_4$, compared with $(TG)_{27}$ in Lake Gjerdhaugvatnet and $(TG)_{31}$ at the marine site. The repeat numbers of the most common allelic variants for each of the five groups of sticklebacks examined are shown in Figure 3. Below is a more detailed assessment of the TG-repeat arrays for all specimens analysed.

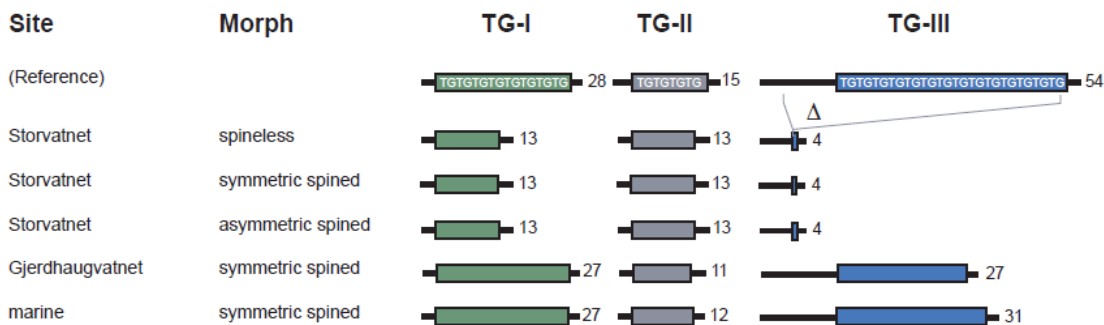

**Figure 3.** Representative examples of allelic variants of *Pel*A caused by variable numbers of TG dinucleotides at TG-repeats I–III. The repeat number of the most common allelic variants is shown for each of the five groups of sticklebacks examined. For more details see the Supplementary Figures S2 and S3.

### 3.2.1. TG-Repeat Array I

At TG-repeat I, the repeat length varied from $(TG)_{13}$ to $(TG)_{32}$ among sticklebacks from the three sites sampled (Figure 4a, Supplementary Figures S2 and S3). In Lake Storvatnet, repeat lengths ranged from $(TG)_{13}$ to $(TG)_{26}$ (Figure 4a, Supplementary Figure S2). $(TG)_{13}$ dominated in this lake with a frequency of 0.87, the presence of at least one copy in all specimens except in one spineless individual, and low heterozygosity (0.21). The array lengths of spineless and symmetrical spined specimens in Lake Storvatnet were within the same interval, from $(TG)_{13}$ to $(TG)_{32}$ (Figure 4a, Supplementary Figure S2a,b), and all asymmetrical specimens had $(TG)_{13}$ (Figure 4a, Supplementary Figure S2c). TG-repeat I among specimens from Lake Gjerdhaugvatnet varied from $(TG)_{26}$ to $(TG)_{30}$ and was relatively uniform although all specimens were heterozygous (Figure 4a, Supplementary Figure S3a). Their conspecifics at the marine site varied more at TG-repeat I, from $(TG)_{13}$ to $(TG)_{32}$ with a heterozygosity of 0.86 (Figure 4a, Supplementary Figure S3b).

### 3.2.2. TG-Repeat Array II

At TG-repeat II, the repeat length varied from $(TG)_9$ to $(TG)_{16}$ among all sticklebacks sampled (Figure 4b, Supplementary Figures S2 and S3). In Lake Storvatnet, repeat lengths ranged from $(TG)_{11}$ to $(TG)_{16}$, and $(TG)_{13}$ was present in all the examined specimens, with an allele frequency of 0.92 (Figure 4b, Supplementary Figure S2). The TG-repeats II of spineless and asymmetric-spined specimens in Lake Storvatnet were within the same interval, from $(TG)_{13}$ to $(TG)_{16}$ repeats, whereas two symmetric individuals were heterozygous $(TG)_{11/13}$ (Figure 4b and Supplementary Figure S2). In Lake Gjerdhaugvatnet, the number of $(TG)_n$ was uniform with all specimens being homozygous for $(TG)_{11}$ (Figure 4b, Supplementary Figure S3), whereas their marine conspecifics varied from $(TG)_9$ to $(TG)_{16}$, with a single heterozygous individual (Figure 4b, Supplementary Figure S3a).

### 3.2.3. TG-Repeat Array III

At TG-repeat III, the number of repeats varied from $(TG)_4$ to $(TG)_{47}$ among sticklebacks from the three sites sampled (Figure 4c). In Lake Storvatnet, $(TG)_n$ varied from $(TG)_4$ to $(TG)_{43}$ (Figure 4c, Supplementary Figure S2). The short $(TG)_4$ dominated with an allele frequency of 0.76 and the presence of at least one copy in each of the spineless (Supplementary Figure S2a), symmetric-spined (Supplementary Figure S2b), and asymmetric-spined (Supplementary Figure S2c) specimens. Spineless, symmetric-spined and asymmetric-spined specimens in Lake Storvatnet had TG-III repeats within the same interval from $(TG)_4$ to approximately $(TG)_{43}$ (Figure 4c, Supplementary Figure S2). In Lake Gjerdhaugvatnet, the number of repeats at TG-repeat III varied from $(TG)_{25}$ to $(TG)_{30}$, and $(TG)_{27}$ was present with at least 1 copy in all but 1 of the 12 examined specimens and with an allele frequency of 0.79 (Figure 4c, Supplementary Figure S3a). Specimens from the

marine site varied from $(TG)_{24}$ to $(TG)_{47}$ and were heterozygous throughout (Figure 4c, Supplementary Figure S3b).

An additional polymorphism was found upstream of and flanking TG-repeat III in specimens from Lake Storvatnet. At this upstream flanking region, all specimens from Lake Storvatnet had a 58 bp deletion compared with the reference (Figure 5; Supplementary Figure S6-3). In contrast, all of the examined specimens from Lake Gjerdhaugvatnet (N = 12) and the marine site (N = 7) conformed to the reference in this respect.

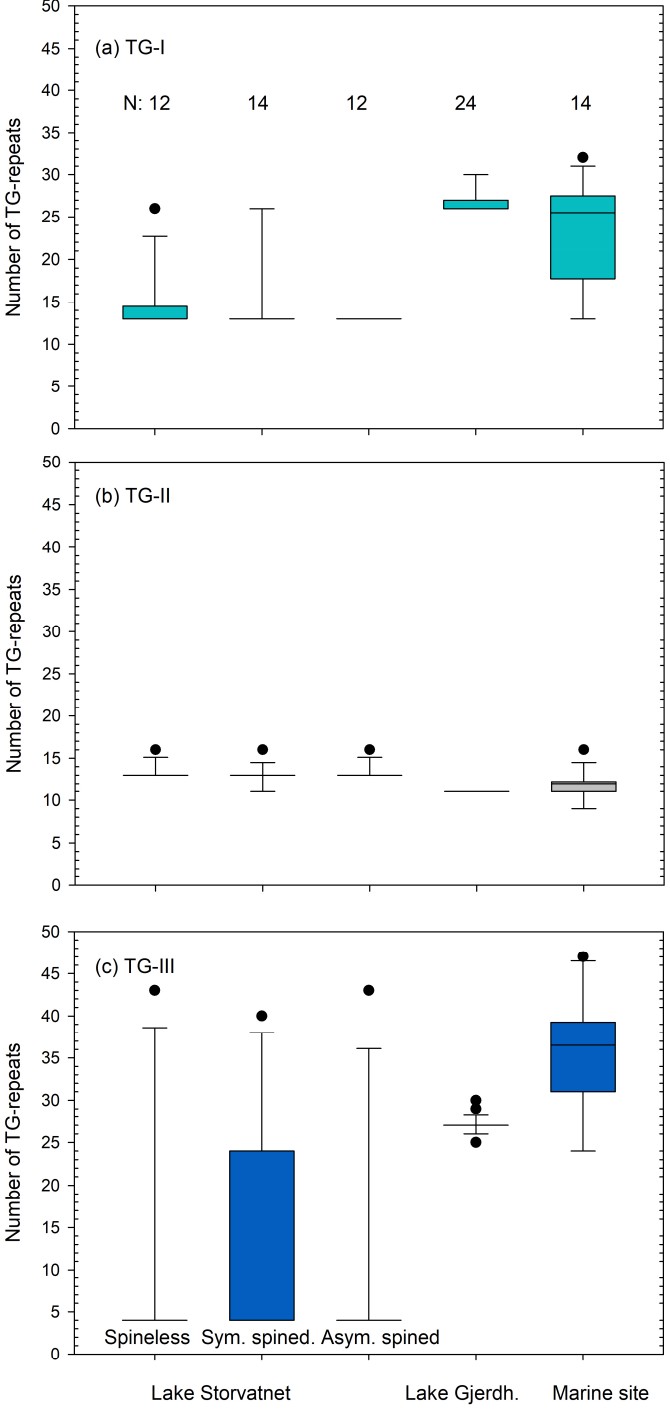

**Figure 4.** Box–whisker plots showing the number of repeats of (**a**) TG-I, (**b**) TG-II, and (**c**) TG-III, for spineless, symmetric, and asymmetric specimens from Lake Storvatnet, and spined specimens from Lake Gjerdhaugvatnet and the marine site. See also Supplementary Figures S2 and S3. Numbers ("N") in the figure show the number of alleles examined.

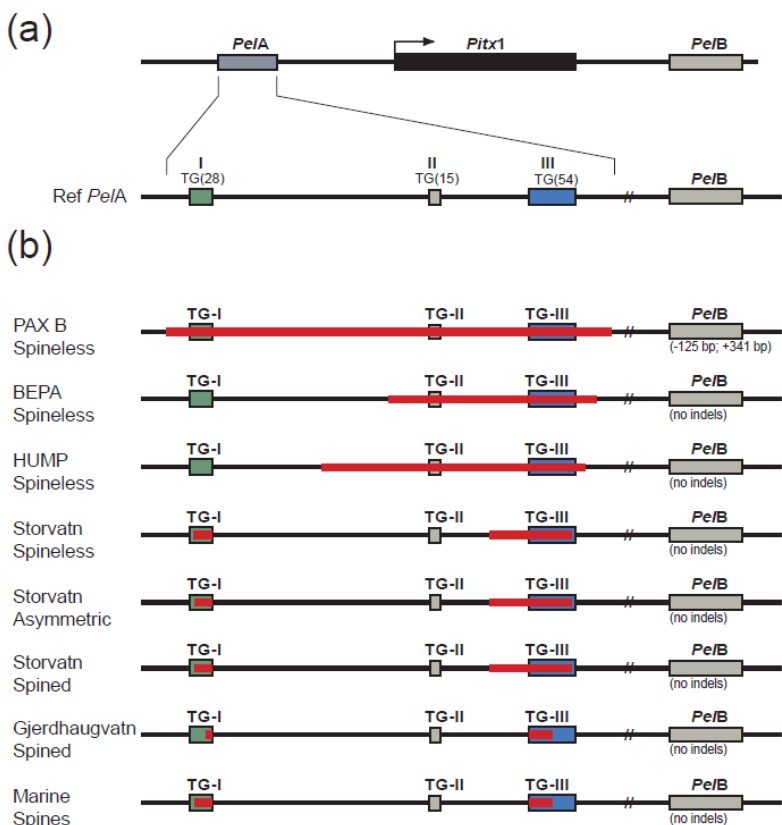

**Figure 5.** (**a**) The relative positions of the *Pitx*1 gene and its upstream and downstream enhancers, *Pel*A and *Pel*B. In addition, the relative positions within *Pel*A of TG-repeat I (green), TG-repeat II (grey), and TG-repeat III (blue) are also shown. The approximate downstream position (out of scale) of *Pel*B (grey) is indicated as well, as are the number of TG-repeats at TG-repeat I, II, and III for the reference sequence (GU130435) from Salmon River, British Columbia. (**b**) Polymorphism at TG-repeat I, II, and III located at *Pel*A, and at *Pel*B in spineless sticklebacks from three previously published studies from Paxton Benthic Lake (PAXB), Bear Paw Lake (BEPA), and Hump Lake (HUMP) from the west coast of North America [28]. Spineless, asymmetric-spined, and symmetric-spined sticklebacks from the present study sites of Lake Storvatnet (Storvatn), Lake Gjerdhaugvatnet (Gjerdhaugvatn), and the marine sample (Marine) in sub-Arctic Norway at the north-east coast of Europe are also shown. Missing regions at *Pel*A compared with the reference sequence are indicated in red.

### 3.2.4. Comparing Haplotypes of Spineless and Spined Sticklebacks from Lake Storvatnet

The combination of Sanger sequencing and fragment analyses enabled the haplotypes of *Pel*A to be inferred for specimens that were either homozygous throughout or heterozygous at one of the TG-repeats only. Three spineless (S30, S31, and S42, Supplementary Figure S2a), two symmetric-spined (S03, S07, Supplementary Figure S2b), and four asymmetric-spined specimens (S23, S33, S34 and S43, Supplementary Figure S2c) were all homozygous with haplotypes $(TG)_{13}$, $(TG)_{13}$, and $(TG)_4$ at TG-repeat I, TG-repeat II, and TG-repeat III, respectively. The same haplotype is also the most frequent in each of the 3 groups making up at least 58%, 36%, and 75% of the haplotypes among spineless, symmetric-spined, and asymmetric-spined individuals, respectively (Supplementary Figure S2a–c).

### 3.3. Allelic Variation of PelB

No indels were revealed by DNA sequencing of the *Pel*B region among the 10, 6, and 5 examined specimens from Lake Storvatnet, Lake Gjerdhaugvatnet, and the marine site, respectively (Figure 5b, Supplementary Figure S6-4). Sequence reads were ambiguous in between two variable poly-G runs (reference 176,958–177,294), but the sequence analyses and gel-based sizing of *Pel*B amplicons from all individuals both supported a lack of sizable

indels within the *Pel*B region (Supplementary Figures S5 and S6-4c). There was no association between pelvic morphs and SNPs upstream of the first poly-G tract (Supplementary Figure S6-4a) and downstream of the last poly-G tract (Supplementary Figure S6-4b).

*3.4. Sequence Alignments of PelA and PelB*

Sequence analyses revealed a number of SNPs in the pelvic enhancer regions. However, there was no apparent association between nucleotide polymorphism and pelvic status among the sticklebacks (Supplementary Figure S6).

## 4. Discussion

A causal connection between deletion mutations affecting the enhancer elements of the *Pitx*1 gene and loss of pelvic spines has been established in several independent stickleback populations in North America, making a strong case for parallel evolution by common molecular pathways [28]. As far as we know, the stickleback population in Lake Storvatnet is unique by its lack of any such obvious relationship between indels mapping to *Pitx*1 enhancer regions and pelvic status, ranging from fully spined via asymmetric spined to spineless.

The present study suggests that presence or absence of pelvic spines in Lake Storvatnet sticklebacks is not explained by the TG-repeat regions TG-I, TG-II, and TG-III only. Firstly, some individuals with and without spines have exactly the same haplotypes at these TG-repeats. Secondly, TG-repeats TG-I, TG-II, and TG-III at the enhancer *Pel*A are within the same length range regardless of spine phenotype. Thirdly, nothing indicates that *Pel*B, or the flanking regions of the TG-repeats I–III at *Pel*A, explains the presence or absence of pelvic spines in sticklebacks from Lake Storvatnet. TG-II and TG-III are located within the 488 bp region of *Pel*A which has previously been reported as lacking in several North American spineless sticklebacks [28]. Spineless fish from Lake Paxton (benthic morph) lack large segments of *Pel*A, including TG-repeats I, II, and III, in addition to indels at *Pel*B (Figure 5). Spine-reduced specimens from Bear Paw Lake and Hump Lake also lack relatively large segments of *Pel*A, which include TG-repeats II and III. Bear Paw Lake and Hump Lake sticklebacks have larger TG-repeat I compared with their conspecifics in Lake Storvatnet. In contrast, TG-repeat II is absent in these two North American lakes and present in Lake Storvatnet (Figure 5).

Paxton sticklebacks, which exhibit the most extensive deletions at *Pel*A, also seem to have the least developed anti-predator defence with respect to the pelvis structure (or pelvic girdle). Approximately 80 per cent of the adult specimens (benthic morph) in Lake Paxton lack the entire pelvic girdle according to [27]. This is high compared with 12.7 and 7.6 per cent that lack the entire pelvic girdle in Hump Lake and Bear Paw Lake, respectively [18], and especially compared with the complete absence of such individuals in Lake Storvatnet. Moreover, the percentage of sticklebacks lacking both pelvic spines (which are part of the pelvic structure/girdle) regardless of the rest of the pelvic girdle is $\geq$ 80, 77, 92, and 30 for Paxton Lake, Hump Lake, Bear Paw Lake [18,27], and Lake Storvatnet, respectively. However, it is premature to draw conclusions about any association between the size of *Pel*A and lack of pelvic spines (and pelvic girdle) based on a few individuals from three North American and one North European stickleback population.

The relationship between the *Pel*A enhancer and presence of pelvic spines was nicely demonstrated by Chan and colleagues [28]. Quantitative traits loci analyses and DNA sequencing studies have also pointed at chromosome 7 close to where *Pitx*1 and *Pel*A are located (see Introduction), as a position of loci coding for pelvic spines. Thus, the lack of any association between the *Pel*A variants and pelvic spine status in Lake Storvatnet is challenging to explain, but other genetic loci have been suggested to be involved in the development of pelvic spines as well. Based on linkage mapping and QTL analysis, additional loci suggested to play a role in the fine-tuning of the length of the pelvic spines (not to be confused with loci coding for presence or absence of pelvic spines) seem to be located at chromosome 2 and 4 [15] and chromosome 8 [29].

The percentage of specimens from Lake Storvatnet with asymmetrical pelvic spines is similar to a previous report from the same lake [43]. The significantly higher number of left- compared with right-biased individuals among these asymmetrical fish concurs with previous reports from a majority of populations of pelvic-reduced sticklebacks in North America (see [44]). Bell and collaborators [44] gave an overview of potential reasons for the asymmetrical pelvic spines and suggested that (i) asymmetry is associated with lack of *Pitx*1 expression, and (ii) *Pitx*2 and some other loci or genetic mechanisms may play a role in the asymmetry as well. Sticklebacks in Lake Storvatnet also seem to have a genetic component in the asymmetry of their pelvic spines. This is suggested by the significantly higher abundance of left- compared with right-biased asymmetric specimens compared with the expected abundance with random asymmetry (50% of each). However, such a genetic component does not exclude random phenotypic variation in symmetry due to developmental instability.

At this point, we can only speculate about the reason for the lack of association between *Pel*A and pelvic spine status among sticklebacks in Lake Storvatnet. Firstly, TG-repeats are known to form left-handed, fragile Z-DNA, which is prone to deletions [38,39]. Z-DNA opens up the chromatin structure which allows transcription factors to bind to the enhancer [38]. Thus, TG-repeats of certain lengths creating left-handed Z-DNA sequences may be required for chromatin-dependent activation of promoters and for transcription to occur [38]. The pelvic enhancers might not function effectively in specimens with large TG-repeat regions deleted, such as in 9 different pelvic-reduced stickleback populations with deleted sequences of from 757 to approximately 5000 bp [28]. The *Pel*A variants among sticklebacks in Lake Storvatnet are also relatively short. Thus, one might speculate that the size of *Pel*A variants in this population are at a tipping point for Z-DNA formation and transcription to occur or not, leaving spined and spine-reduced individuals to develop based on additional genetic factors, epistatic and epigenetic effects, and/or environmental cues.

The $Ca^{2+}$ concentration in Lake Storvatnet (0.9 mg/L) is well within the range of 0.07–13 mg/L from 1000 Norwegian freshwater lakes reported by [45]. On the other hand, the $Ca^{2+}$ concentration in Lake Storvatnet is relatively low compared with three Norwegian freshwater lakes inhabited by spineless sticklebacks with 5.5 mg/L and 3.0 mg/L [16] and 7.9 mg/L (unpubl. data, J.T. Nordeide). Thus, the low $Ca^{2+}$ concentration in Lake Storvatnet may be interpreted as strengthening the tipping point hypothesis (above).

An alternative, though not mutually exclusive, explanation for the lack of association between the different variants of *Pel*A and pelvic spine status in Lake Storvatnet has to do with reduced standing genetic variation and subsequent alternative genetic pathways to adapt to freshwater environments. Fang and collaborators [46] suggested that contemporary threespine stickleback populations originated in the Eastern Pacific Ocean and North America, while some sticklebacks subsequently migrated to colonize other regions including the Atlantic Ocean and Northern Europe. Thus, the ancestral populations from the Eastern Pacific region have a higher standing genetic diversity than stickleback populations from other geographical regions [17,46,47]. Such inter-regional differences in standing genetic variation have been suggested to give striking differences in the proportion of loci involved in freshwater adaptations along the west coast of North America and Northern Europe [17,47]. Moreover, Kemppainen and colleagues [48] advocated that *Pitx*1's role in coding for pelvic spines of pelvic-reduced nine-spined sticklebacks (*Pungitius pungitius*) has been replaced by alternative loci in some North European populations. Pelvic spines in these populations were suggested to be a polygenic trait coded for by loci located near 10 novel QTLs [48]. At the moment we can only speculate whether other loci than *Pel*A and *Pel*B take part in controlling the expression of pelvic spines in some threespine stickleback populations as well, such as the one in Lake Storvatnet. Future whole-genome sequencing of the different polymorphic forms of sticklebacks in Lake Storvatnet and examination of population genetic parameters for genetic diversity and differentiation might contribute to locate alternative loci controlling the expression of pelvic spines (see [49,50]).

The presence of spined, spineless, and asymmetric specimens from Lake Storvatnet, and the lack of spineless fish from the downstream Lake Gjerdhaugvatnet and from the marine site, concurs with results from previous studies of spine morphology from the same sites [16,43]. Comparison of *Pel*A variants between the three sites in the present study revealed a few trends (Figure 3, Figure 4, Supplementary Figures S2 and S3). First, the diversity at *Pel*A of the relatively few specimens examined seems high in the marine threespine sticklebacks compared with those from Storvatnet, and those from Lake Gjerdhaugvatnet in particular. This is as expected according to the founder effect and the putatively larger effective population size of sticklebacks in the sea. Second, *Pel*A variants were in general shorter among Lake Storvatnet sticklebacks than in the two downstream populations, especially due to TG-repeat III. TG-repeat II was of approximately the same length in all three populations, whereas at TG-repeat I, the specimens in Lake Gjerdhaugvatnet have relatively uniform and long TG-repeat sequences.

## 5. Conclusions

Lake Storvatnet sticklebacks carry unique variants of the *Pel*A enhancer region. No simple association was detected between the pelvic spine status and *Pel*A among sticklebacks from Lake Storvatnet. The *Pel*A enhancers of sticklebacks from Lake Storvatnet were short compared with their spined conspecifics in the downstream Lake Gjerdhaugvatnet and the nearby marine site, yet they were relatively long compared with those of pelvic-spine-reduced threespine sticklebacks from three North American populations. No polymorphism was found at *Pel*B. These results clearly indicate that there are alternative molecular pathways to parallel evolution of pelvic reduction in threespine sticklebacks, which could include epistatic and epigenetic effects, and/or environmental cues.

**Supplementary Materials:** The following supporting information can be downloaded at: https://www.mdpi.com/article/10.3390/fishes8030164/s1, Supplementary Figure S1: Drawings of the ventral part of fully spined and spine-reduced threespine sticklebacks; Supplementary Figure S2: Number of thymine-guanine repeats [$(TG)_n$] at the enhancer *Pel*A of spined and spine-reduced specimens from Lake Storvatnet; Supplementary Figure S3: Number of thymine-guanine repeats [$(TG)_n$] at the enhancer *Pel*A of fully spined specimens from Lake Gjerdhaugvatnet and a nearby marine site; Supplementary Figure S4: DNA sequencing of *Pel*A enhancers from threespine sticklebacks and a reference sequence. Primers used in the present study are also shown; Supplementary Figure S5: DNA sequencing of *Pel*B enhancers from threespine sticklebacks and a reference sequence. Primers used in the present study are shown; Supplementary Figure S6: DNA sequence alignments upstream and downstream of TG-repeats I–III of *Pel*A, and *Pel*B sequences of specimens from Lake Storvatnet, Lake Gjerdhaugvatnet, and the marine site.

**Author Contributions:** Conceptualization, D.A., S.D.J., T.B.M., and J.T.N.; methodology, D.A., S.D.J., T.B.M., and J.T.N.; acquisition, all authors; lab work, D.A. and I.K.H.; validation, all authors; formal analysis, all authors; investigation, all authors; resources, S.D.J., T.B.M., and J.T.N.; data curation, D.A.; writing—original draft preparation, all authors; visualization, D.A. and S.D.J.; supervision, S.D.J., T.B.M., and J.T.N.; project administration, J.T.N.; funding acquisition, J.T.N. All authors have read and agreed to the published version of the manuscript.

**Funding:** This research received no external funding.

**Institutional Review Board Statement:** Ethical review and approval were waived for this study due to the study being carried out according to ethical guidelines stated by the Norwegian Ministry of Agriculture and Food through the Animal Welfare Act. According to these guidelines, we were not required to—and therefore do not—have a specific approval or approval number.

**Informed Consent Statement:** Not applicable.

**Conflicts of Interest:** The authors declare no conflict of interest.

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
