# Peer review of "Pitx1 Enhancer Variants in Spined and Spine-Reduced Subarctic European Sticklebacks"

_fishes, doi:10.3390/fishes8030164_

Round 1

Reviewer 1 Report

The article reported phenotypic variation of pelvic spines and the molecular variation at PelA and PelB in a comparison between (i) spined and spine-reduced sticklebacks within Lake Storvatnet, and (ii) Lake Storvatnet sticklebacks and (spined) conspecifics from the downstream Lake Gjerdhaugvatnet and marine specimens.

Although this is a quite interesting topic, but the manuscript needs major revision before acceptance. I will list a few comments that support my position.

1.      The introduction should be focused and brief because it is currently too long and confusing.

2.      For the reader's better understanding, include a GIS map showing all the locations of the sampling sites. The GIS location map (a) and schematic diagram (b) can be combined into a single figure.

3.      For morphological studies sample size is adequate. However, the sample size is quite small for molecular investigations.

4.      I strongly believe that the sample size used for SNP detection in the present study is very low. The SNP detection is stringency was not enough (NGS is better approach than the Sanger sequencing). 

Additionally, SNP validation must be done, for instance via qPCR analysis.

Therefore, I recommend leaving the SNP part out of the current manuscript.

5.      The Ca2+ content is the only water quality metric stated by the authors. Additionally, they had no correlation with any other factors (morphological and molecular). For a better understanding of the function of the ecosystem, I think it would have been better to measure and correlate with more metrics related to the quality of the water and sediment.

6.      The study includes a major lacuna since the authors speculated that other molecular pathways may be involved but were unable to define the causes of the evolution of pelvic reduction in three-spine sticklebacks. The discussion is poorly developed and conclusions are vague.

7.      However, the study showed that sticklebacks in Lake Storvatnet are distinct and that there may be some environmental or evolutionary impact. Phenotypic plasticity and the involvement of epigenetics play a key role in the species' ability to adapt to evolutionary or environmental stresses. Therefore, to improve the MS, include this information in the Discussion & Conclusion part.

Reviewer 2 Report

fishes-2180076

Pitx1 enhancer variants in spined and spine-reduced subarctic European sticklebacks

The authors investigated the genetic mechanism leading to pelvic spine reduction in sticklebacks. The data are useful, and the paper is well-written overall. I recommend publication with minor revisions. The following comments should be addressed by the authors:

Abstract, line 11: Change “molecular variation” to “sequence variation”.

Methods, 2.2: Please add definitions for pelvic scores 1, 2, and 3.

Table 1b: Define “S.N”.

Table 1b: For consistency, change the Remarks in row 1 from “Spined (Normal)” to “Symmetric Spined”.

Table 1, Note: I suggest rewriting the note: “Note: N (Table 1a) is the total number of specimens from each location; n (Table 1b) is the number of each morphological variant from each location.”

Figure 3: The ranges seem to be missing for the numbers of repeats.

Figure 4: Please add the number of fish sequenced in each category. Also, the legend says “Numbers in the figure show number of alleles examined.” What numbers does that refer to, and is that the same as the number of individual fish?

Figure 5: The font is too small on the sequence labels.

Discussion: The 59 bp deletion and/or the reduction in repeat number at TG-III may be permissive but not sufficient for development of the spineless phenotype.

S4: What does the green, blue, and gray highlighting indicate? Please bold all the primer sequences. The blue and red text do not show up well, I’d suggest underlining all three TG repeats.

Author Response

Reviewer 2

Open Review

English language and style

( ) English very difficult to understand/incomprehensible
( ) Extensive editing of English language and style required
( ) Moderate English changes required
(x) English language and style are fine/minor spell check required
( ) I don't feel qualified to judge about the English language and style

Yes

Can be improved

Must be improved

Not applicable

Does the introduction provide sufficient background and include all relevant references?

(x)

( )

( )

( )

Are all the cited references relevant to the research?

(x)

( )

( )

( )

Is the research design appropriate?

(x)

( )

( )

( )

Are the methods adequately described?

(x)

( )

( )

( )

Are the results clearly presented?

(x)

( )

( )

( )

Are the conclusions supported by the results?

(x)

( )

( )

( )

Comments and Suggestions for Authors

fishes-2180076

Pitx1 enhancer variants in spined and spine-reduced subarctic European sticklebacks

 The authors investigated the genetic mechanism leading to pelvic spine reduction in sticklebacks. The data are useful, and the paper is well-written overall. I recommend publication with minor revisions. The following comments should be addressed by the authors:

 Abstract, line 11: Change “molecular variation” to “sequence variation”. Authors’ reply: Done.

 Methods, 2.2: Please add definitions for pelvic scores 1, 2, and 3. Authors’ reply: A more detailed explanation of the pelvic scores from 0 to 4 is now added in lines 156-162. This explanation comes in addition to the description already given in Supplementary S2.

 Table 1b: Define “S.N”. Authors’ reply: S.N shows the numbers we allocated to each of the specimens. We have removed “S.N.” from the table, since it may be confusing.

Table 1b: For consistency, change the Remarks in row 1 from “Spined (Normal)” to “Symmetric Spined”. Authors’ reply: Changed

Table 1, Note: I suggest rewriting the note: “Note: N (Table 1a) is the total number of specimens from each location; n (Table 1b) is the number of each morphological variant from each location.” Authors’ reply: Changed.

Figure 3: The ranges seem to be missing for the numbers of repeats. Authors’ reply: Figure 1 was never supposed to show all details and variety of the TG-repeats. Our intension was to show representative alleles at the three loci. We have made this clearer in the legend to Fig. 3 and we have included also in the figure legend that more details can be found in Suppl. S2 and S3.

Figure 4: Please add the number of fish sequenced in each category. Also, the legend says “Numbers in the figure show number of alleles examined.” What numbers does that refer to, and is that the same as the number of individual fish? Authors’ reply: The reviewer is correct. Unfortunately, these numbers disappeared during the preparation or uploading of the manuscript, for some unknown technical reason. We have now added these numbers and they show number of alleles examined. Stickleback is a diploid species which means that the number of individuals examined is half the number of alleles.

Figure 5: The font is too small on the sequence labels. Authors’ reply: The fonts are now bigger.

Discussion: The 59 bp deletion and/or the reduction in repeat number at TG-III may be permissive but not sufficient for development of the spineless phenotype. Authors’ reply: This is a good point which we addressed on lines 332-347 and 380-394 . However, these indels per se cannot explain why some individuals are spined while others are spineless, since there are no differences between the symmetrical spined, asymmetrical spined and spineless specimens.

S4: What does the green, blue, and gray highlighting indicate? Please bold all the primer sequences. The blue and red text do not show up well, I’d suggest underlining all three TG repeats. Authors’ reply: The highlights are removed or changed according to the suggestions. See Suppl. S4.

Reviewer 3 Report

In this paper, the authors studied the relationship between patterns of molecular variation of pelvic enhancers in sticklebacks collected in subarctic Norway, with their morphological characteristics describing the pelvis. It studied both inter-population, and within-population variation, and most attention has been paid to the latter. No relationship between genetic and morphological characteristics, including the population in lake Storvatnet where the morphological variation occurs (this is not the case for the other two locations), was found. The authors consider this situation (Lake Storvatnet) to be unique because, in a number of other similar cases in other studies, the association between morphological and genetic characteristics has been found.

I think that this study is an important contribution to the area of population genetics and morphology and is deserved to be published in Fishes, although I have some comments regarding the interpretation of obtained interesting results. The morphological results obtained are closely dealing with asymmetry, but this part of the study is practically not developed and is not theoretically conceptualized, although may be directly linked to interpretations. The authors do not identify the type of asymmetry they are dealing with – it can be directional asymmetry, antisymmetry, fluctuating asymmetry, and their combination (see, for instance, Graham, J.H.; Raz, S.; Hel-Or, H.; Nevo, E. Fluctuating asymmetry: methods, theory, and applications. Symmetry 2010, 2, 466–540). This is important because fluctuating asymmetry has no genetic basis whereas two other types of asymmetry have. From table 1 and figure S2, it can be assumed, that the pelvic structures manifest fluctuating asymmetry, and, maybe, directional asymmetry, but it is not clear if the latter is statistically significant or not. Presence, and seemingly, a high contribution of fluctuating asymmetry into the total phenotypic variance of the studied structure, suggests that a significant portion of within-population phenotypic variation may be caused by developmental instability, which practically does not have a genetic basis (see Graham, J.H. Nature, nurture, and noise: developmental instability, fluctuating asymmetry, and the causes of phenotypic variation. Symmetry 2021b, 13, 1204 https://doi.org/10.3390/sym13071204, and Lajus, D. God playing dice, revisited: determinism and indeterminism in studies of stochastic phenotypic variation. Emerg. Top. Life Sci. 2022. 6(3): 303-310. DOI: 10.1042/ETLS20210285).

It means that the observed absence of association between genetic and morphological characteristics may be observed not only because the relevant genetic characteristics are not found so far, but also because morphological traits may have the essential portion of variation which does not have a genetic basis being determined by developmental instability. Provided data that allow assuming quite a high contribution of fluctuating asymmetry into the variation of pelvic structures, make such interpretation quite probable. 

Author Response

Reviewer 3

Open Review

English language and style

( ) English very difficult to understand/incomprehensible
( ) Extensive editing of English language and style required
( ) Moderate English changes required
( ) English language and style are fine/minor spell check required
(x) I don't feel qualified to judge about the English language and style

Yes

Can be improved

Must be improved

Not applicable

Does the introduction provide sufficient background and include all relevant references?

( )

(x)

( )

( )

Are all the cited references relevant to the research?

(x)

( )

( )

( )

Is the research design appropriate?

(x)

( )

( )

( )

Are the methods adequately described?

(x)

( )

( )

( )

Are the results clearly presented?

( )

( )

(x)

( )

Are the conclusions supported by the results?

( )

( )

(x)

( )

Comments and Suggestions for Authors

In this paper, the authors studied the relationship between patterns of molecular variation of pelvic enhancers in sticklebacks collected in subarctic Norway, with their morphological characteristics describing the pelvis. It studied both inter-population, and within-population variation, and most attention has been paid to the latter. No relationship between genetic and morphological characteristics, including the population in lake Storvatnet where the morphological variation occurs (this is not the case for the other two locations), was found. The authors consider this situation (Lake Storvatnet) to be unique because, in a number of other similar cases in other studies, the association between morphological and genetic characteristics has been found.

I think that this study is an important contribution to the area of population genetics and morphology and is deserved to be published in Fishes, although I have some comments regarding the interpretation of obtained interesting results. The morphological results obtained are closely dealing with asymmetry, but this part of the study is practically not developed and is not theoretically conceptualized, although may be directly linked to interpretations.

The authors do not identify the type of asymmetry they are dealing with – it can be directional asymmetry, antisymmetry, fluctuating asymmetry, and their combination (see, for instance, Graham, J.H.; Raz, S.; Hel-Or, H.; Nevo, E. Fluctuating asymmetry: methods, theory, and applications. Symmetry 2010, 2, 466–540). This is important because fluctuating asymmetry has no genetic basis whereas two other types of asymmetry have. From table 1 and figure S2, it can be assumed, that the pelvic structures manifest fluctuating asymmetry, and, maybe, directional asymmetry, but it is not clear if the latter is statistically significant or not. Presence, and seemingly, a high contribution of fluctuating asymmetry into the total phenotypic variance of the studied structure, suggests that a significant portion of within-population phenotypic variation may be caused by developmental instability, which practically does not have a genetic basis (see Graham, J.H. Nature, nurture, and noise: developmental instability, fluctuating asymmetry, and the causes of phenotypic variation. Symmetry 2021b, 13, 1204 https://doi.org/10.3390/sym13071204, and Lajus, D. God playing dice, revisited: determinism and indeterminism in studies of stochastic phenotypic variation. Emerg. Top. Life Sci. 2022. 6(3): 303-310. DOI: 10.1042/ETLS20210285).

It means that the observed absence of association between genetic and morphological characteristics may be observed not only because the relevant genetic characteristics are not found so far, but also because morphological traits may have the essential portion of variation which does not have a genetic basis being determined by developmental instability. Provided data that allow assuming quite a high contribution of fluctuating asymmetry into the variation of pelvic structures, make such interpretation quite probable. 

Authors’ reply: This is an interesting point from Reviewer 3, although it draws the reader’s attention away from the main aim of our study. However, we have added a paragraph about this asymmetry in the Discussion (lines 418-428). This is a good topic for further studies, but beyond the scope of this report.

Round 2

Reviewer 3 Report

I am glad that the authors consider my comment “a good topic for further studies” and also made an addition to the manuscript. I do not think, however, that this effectively responses to my comment.

In the added fragment, the authors argue that the asymmetry observed in their study is left-biased and similar to previous studies. And later they write, “Thus, asymmetrical pelvic spines in sticklebacks seems to have a genetic basis”. However, I do not see in the manuscript any statistical proof that the observed asymmetry is directional, i.e. heritable. The similarity of the obtained results with the literature data cannot be considered as such proof. Also, the authors conclude that the observed asymmetry is “probably not due to environmental stress and developmental instability alone”. Here I would like to note that environmental stress and developmental instability are different factors because environmental stress is one of factors that influences developmental instability.

A very brief and superficial analysis of Table 1 shows that almost 100 individuals are asymmetrical, and if to assume that this asymmetry is fluctuating (as the opposite is not proved), i.e. combination of left and right values of the trait is fully random, 50 individuals should have no spines at all, and 50 should have both left and right spines. This model describes the situation when a variation in the number of spines is by developmental instability alone. In this case correlation between morphological and genetic (or any other type of parameters) is impossible. One can see that this model explains a significant part of observed (Table 1) variation, yet not all variation. There is an excessive number of symmetrical individuals without spines (92-50=42) and with two spines (113-50=63), i.e. only about 2/3 of all individuals are described by the model, which means the presence of some genetic base of morphological variation of this trait. But still, an essential part of the variation of this trait is random, and thus it essentially reduces the probability to find the correlation between genetic and morphological traits, which is one of the main points of discussion of this paper. If the presence of directional asymmetry is proved, the proportion of random variation is lower.

I, however, do not insist to include these analyses in the manuscript. As a minimum requirement, I would recommend performing statistical tests on the presence of directional asymmetry in variation number spines, and to mention the presence of random phenotypic variation, in this trait, which may seriously confound the correlation between morphological and genetic traits.

Author Response

Dear reviewers and editors

We have tried to include the information as requested by reviewer 3. See our replies below.

Open Review

Quality of English Language

( ) English very difficult to understand/incomprehensible
( ) Extensive editing of English language and style required
( ) Moderate English changes required
( ) English language and style are fine/minor spell check required
(x) I am not qualified to assess the quality of English in this paper

Comments and Suggestions for Authors

I am glad that the authors consider my comment “a good topic for further studies” and also made an addition to the manuscript. I do not think, however, that this effectively responses to my comment.

In the added fragment, the authors argue that the asymmetry observed in their study is left-biased and similar to previous studies. And later they write, “Thus, asymmetrical pelvic spines in sticklebacks seems to have a genetic basis”. However, I do not see in the manuscript any statistical proof that the observed asymmetry is directional, i.e. heritable.

Authors’ reply: We have included statistical tests which show that the number of left- and right-biased fish is significantly different from the ratio 1:1 which is expected if random events caused the asymmetry (lines 261-264).

The similarity of the obtained results with the literature data cannot be considered as such proof. Also, the authors conclude that the observed asymmetry is “probably not due to environmental stress and developmental instability alone”. Here I would like to note that environmental stress and developmental instability are different factors because environmental stress is one of factors that influences developmental instability.

Authors’ reply: This is what we meant as well in the previous version. We have now changed this sentence (lines 424-435). Hopefully this reads better now.

A very brief and superficial analysis of Table 1 shows that almost 100 individuals are asymmetrical, and if to assume that this asymmetry is fluctuating (as the opposite is not proved), i.e. combination of left and right values of the trait is fully random, 50 individuals should have no spines at all, and 50 should have both left and right spines. This model describes the situation when a variation in the number of spines is by developmental instability alone. In this case correlation between morphological and genetic (or any other type of parameters) is impossible. One can see that this model explains a significant part of observed (Table 1) variation, yet not all variation. There is an excessive number of symmetrical individuals without spines (92-50=42) and with two spines (113-50=63), i.e. only about 2/3 of all individuals are described by the model, which means the presence of some genetic base of morphological variation of this trait. But still, an essential part of the variation of this trait is random, and thus it essentially reduces the probability to find the correlation between genetic and morphological traits, which is one of the main points of discussion of this paper. If the presence of directional asymmetry is proved, the proportion of random variation is lower.

Authors’ reply: If we understand the reviewer correctly this has been corrected now (lines 424-435).

 I, however, do not insist to include these analyses in the manuscript. As a minimum requirement, I would recommend performing statistical tests on the presence of directional asymmetry in variation number spines, and to mention the presence of random phenotypic variation, in this trait, which may seriously confound the correlation between morphological and genetic traits.

Authors’ reply: Hopefully we have interpreted reviewer 3’s suggestions correctly and carried out the changes accordingly in lines 262-266 and lines 424-435.